# Multisensor and Multiscale Data Integration Method of TLS and GPR for Three-Dimensional Detailed Virtual Reconstruction

**DOI:** 10.3390/s23249826

**Published:** 2023-12-14

**Authors:** Di Zhang, Dinghan Jia, Lili Ren, Jiacun Li, Yan Lu, Haiwei Xu

**Affiliations:** 1College of Civil Engineering, Henan University of Engineering, Zhengzhou 451159, China; 202010816227@stu.haue.edu.cn (D.J.); luyan@haue.edu.cn (Y.L.); haiweixu@haue.edu.cn (H.X.); 2College of Resource Environment and Tourism, Capital Normal University, Beijing 100048, China; lijiacun@cnu.edu.cn; 3Henan Shijiguoke Aerospace Technology Co., Ltd., Zhengzhou 450008, China; renll@21at.com.cn

**Keywords:** TLS, GPR, data fusion, time synchronization algorithm, coordinate transformation

## Abstract

Integrated TLS and GPR data can provide multisensor and multiscale spatial data for the comprehensive identification and analysis of surficial and subsurface information, but a reliable systematic methodology associated with data integration of TLS and GPR is still scarce. The aim of this research is to develop a methodology for the data integration of TLS and GPR for detailed, three-dimensional (3D) virtual reconstruction. GPR data and high-precision geographical coordinates at the centimeter level were simultaneously gathered using the GPR system and the Global Navigation Satellite System (GNSS) signal receiver. A time synchronization algorithm was proposed to combine each trace of the GPR data with its position information. In view of the improved propagation model of electromagnetic waves, the GPR data were transformed into dense point clouds in the geodetic coordinate system. Finally, the TLS-based and GPR-derived point clouds were merged into a single point cloud dataset using coordinate transformation. In addition, TLS and GPR (250 MHz and 500 MHz antenna) surveys were conducted in the Litang fault to assess the feasibility and overall accuracy of the proposed methodology. The 3D realistic surface and subsurface geometry of the fault scarp were displayed using the integration data of TLS and GPR. A total of 40 common points between the TLS-based and GPR-derived point clouds were implemented to assess the data fusion accuracy. The difference values in the x and y directions were relatively stable within 2 cm, while the difference values in the z direction had an abrupt fluctuation and the maximum values could be up to 5 cm. The standard deviations (STD) of the common points between the TLS-based and GPR-derived point clouds were 0.9 cm, 0.8 cm, and 2.9 cm. Based on the difference values and the STD in the x, y, and z directions, the field experimental results demonstrate that the GPR-derived point clouds exhibit good consistency with the TLS-based point clouds. Furthermore, this study offers a good future prospect for the integration method of TLS and GPR for comprehensive interpretation and analysis of the surficial and subsurface information in many fields, such as archaeology, urban infrastructure detection, geological investigation, and other fields.

## 1. Introduction

In contrast with traditional methods, TLS is generally used to acquire detailed spatial information for reconstructing 3D realistic models because it offers high-resolution dense point clouds with highly automated and non-contacted operation in a short time, such as in the geosciences [1,2,3,4], architecture [5,6,7], archaeology [8,9], agriculture and forestry [10], engineering, and construction industries [11,12]. This technique generalizes dense point clouds with the position information, reflection intensity, and color information that is used for the visualization and quantitative analysis of 3D realistic models. GPR has been commonly chosen to delineate the shallow geometry of subsurface materials or buried objects in a nondestructive and cost-effective manner. This technology has been widely applied in geology investigations [13,14,15,16], cultural heritage protection [9,17,18,19], urban infrastructure detection [20,21,22,23], water dynamics investigation [24], etc. Owing to the relative electrical property contrasts in the different media, the shallow geometry of subsurface materials or buried objects was revealed on a gray or color two-dimensional (2D) GPR profile. Although the TLS is capable of recording the detailed 3D geometric information of real-world objects or environments, it is challenging to characterize the shallow geometry of subsurface materials with nonintrusive intervention. In addition, the interpretation and understanding of GPR data are unreadable because of the remarkable variation in the radar waveform and relative amplitude. Without corresponding superficial data or known underground structures, the GPR interpreted results are uncertain. Compared with single TLS or GPR data, fusion data permit multisensor and multiscale spatial data for interpreting and analyzing surficial and subsurface information. Moreover, the shallower buried structures obtained by GPR will be better understood with the help of its corresponding superficial data. As a result, data fusion of TLS and GPR was fundamental for comprehensive interpretation and analysis of the surficial and subsurface information. It is extremely valuable to use the multisensor and multiscale data integration method of TLS and GPR for 3D detailed virtual reconstruction.

In recent years, the integration of TLS and GPR has been gradually used in cultural heritage protection [9,12,17,19,25,26], urban infrastructure detection [27,28,29], bridge structure assessment [8,30,31,32], geology investigations [33,34,35,36], glacier detection [37], and other applications. Spahic et al. [38] rebuilt a 3D model of fault outcrop in the Gocad v2009 software using T-LiDAR point clouds and GPR images. Bubek et al. [34] combined TLS topography and 2D GPR images to determine the boundary between the hanging wall and the footwall of a normal fault. Dlesk [39] integrated TLS and GPR data in Rhinoceros 6.0 3D software for 3D documentation of historical tunnels. Ercoli et al. [40] combined the TLS point clouds and 2D GPR images to investigate the ancient polygonal walls of Amelia. Cowie et al. [41] constrained the location of a fault plane based on the TLS and GPR data to obtain the vertical displacement of the normal fault. Puente et al. [9] presented a 3D reconstruction of the Roman site “Aquis Querquennis” using fusion data of GPR, T-LiDAR, and IRT, and the T-LiDAR orthoimages were combined with GPR time slices for comprehensive interpretation. Zhang et al. [42] provided data visualization of fault scarps in Bentley Pointools 1.5 software using TLS point clouds and GPR data. Zhang et al. [36] used TLS-derived data and 2D GPR images to reconstruct a 3D surficial and subsurface model of the Yushu fault. Previous research on the data fusion method of TLS and GPR mainly focused on two aspects. First, the 2D GPR images were directly overlaid on the TLS-based point clouds using computer vision theory [34,39,40,41]. This has the advantage of quickly realizing the data integration of TLS and GPR with different data formats, but the multiperspective spatial data were only displayed in the same scene. Second, the GPR data were transformed to scattered point clouds; then, the point cloud and GPR data were merged into a single point cloud dataset for data visualization [9,36,38,42]. Although a limited number of studies have presented cases for data integration of TLS and GPR in the scattered point cloud form, a reliable systematic methodology is still scarce. Most of these integrated methods were initially realized in the scattered point cloud form by the ground control points (GCPs); therefore, the accuracy of data fusion cannot be ensured. Furthermore, the GPR-derived point clouds with high position information are essential for the data fusion of TLS and GPR. Considering GPR data acquisition and formats, it is difficult to combine each trace of the GPR profile with its position information acquired by the differential GPS (DGPS).

The aim of this study is to develop a reliable systematic methodology for the data integration of TLS and GPR for 3D detailed virtual reconstruction. The GPR data with high-resolution position information at the centimeter level were simultaneously gathered using the pulse signals of the survey wheel. Additionally, a time synchronization algorithm was used to combine the position information of DGPS with each trace of the GPR data. Based on the improved mathematical model of electromagnetic waves, the GPR data were transformed into discrete point clouds in the geodetic coordinate system. In this case, the TLS-based and GPR-derived point clouds were merged into a single point cloud dataset for 3D detailed virtual reconstruction. TLS and GPR (250 MHz and 500 MHz antenna) surveys were conducted in the Litang fault to assess the feasibility and overall accuracy of the proposed methodology.

## 2. The Principle of Integrated TLS and GPR Method

### 2.1. TLS

TLS is a powerful remote sensing technique to capture detailed 3D structures of real-world objects or environments without disrupting said structures. It mainly includes the ranging system, rotating platform, control software, and other accessories. The ranging system is crucial for obtaining the distance between the geometric center of a laser scanner and an object. When the TLS starts to work, the laser scanner emits thousands of pulses to a scanned area with known inclination angles in the horizontal and vertical directions. When a pulse encounters the surface of an object, it will return to the receiver; then, the time of flight and the deflection angle of an emitted pulse are recorded, as shown in Figure 1. Once the time of flight (*t*) is determined, the distance (*D*) between the geometric center of laser scanner and an object can be calculated by Equation (1).
(1)$$D=\frac{1}{2}ct$$
where c represents the speed of light in the vacuum.

Considering the horizontal angle $\left(\alpha \right)$ and the vertical angle $\left(\beta \right)$ of a pulse, the Cartesian coordinates $\left(X_{p},Y_{p},Z_{p}\right)$ of a reflection point can be determined using Equation (2) in the instrument coordinate system, in which the origin of the coordinate system is the geometric center of the prism (Figure 1).
(2)$$\left\{\begin{array}{l} X_{P}=D\cos\beta \cos\alpha \\ Y_{P}=D\cos\beta \sin\alpha \\ Z_{P}=D\sin\beta \end{array}\right.$$

**Figure 1 sensors-23-09826-f001:**
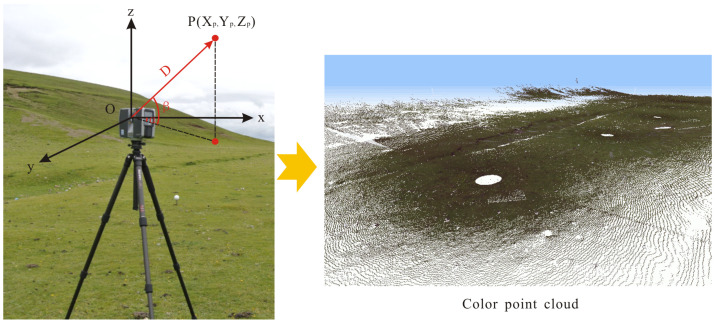
Working principle of TLS. O-xyz represents the instrument coordinate system of TLS.

Point clouds obtained by the TLS are a high-resolution dataset and include the x, y, and z coordinates of the scanned scene; the intensity value; the RGB information; etc. (Figure 1). Color texture maps can be acquired by high-resolution camera into the scanner or externally, which provides the RGB value to the color point clouds. In addition, point clouds in the instrument coordinate system can be transformed into the geodetic coordinate system using the GCPs.

### 2.2. GPR

GPR is a nondestructive geophysical technique to image the shallower buried structures using high-frequency electromagnetic waves [43,44]. Subsurface structures on the GPR data are interpreted by changes in the radar waveform and relative amplitude. As shown in Figure 2, when the GPR antenna is moved, the transmitting antenna emits a series of electromagnetic waves into the ground. Once electromagnetic pulses encounter the material’s changes in different dielectric and electric properties, some of the electromagnetic energy will return to the receiving antenna and other portions will transmit into deeper layers. The reflection electromagnetic signals are amplified and displayed on the screen with a single-channel reflection wave. As the GPR antenna is moved along the survey lines, a serial of sequential single-channel reflection waves are collected and displayed on a gray or color 2D radargram, which is an XZ profile. In view of the working principle, the spatial coordinate system of the GPR data can be set up as shown in Figure 2 [45]. The x axis shows the antenna’s displacement along the GPR measurement lines, the y axis represents the number of the GPR survey lines, and the z axis indicates the two-way travel time of the radar reflection waves. If the velocity of electromagnetic waves in the underground media is known, the depth position of electromagnetic waves can be determined using time–depth conversion [43].

The geometry and characteristic of buried materials are identified by analyzing the relative amplitude variation of electromagnetic waves on the GPR image. Generally, the antenna frequency and dielectric constant of subsurface materials (such as grain type distribution, porosity, water content, etc.) are essential for determining the propagation capacity of electromagnetic waves. Electromagnetic waves energy will reduce and attenuate quickly in the subsurface material with a high dielectric constant, but a greater penetration depth will be realized in the subsurface material with a low dielectric constant. As a result, a high-frequency antenna will obtain higher-resolution GPR data than a lower-frequency antenna, but achieve a lower penetration depth in the same subsurface materials.

### 2.3. Data Integration Principle of TLS and GPR

Point clouds, obtained by the TLS, are generally used to show the realistic visualization of real-world objects or environments and involve the Cartesian coordinates (x, y, z) and point attributes (such as intensity value and RGB value). GPR was used to delineate the shallow geometry of buried materials or objects using a gray or color 2D profile, which was composed of the sequential single-channel reflection waves. Shallower buried structures or objects were identified and interpreted on the GPR image by analyzing the relative amplitude variation of electromagnetic waves. Considering the data acquisition and formats of TLS and GPR, it is obvious that the TLS-based point clouds are unable to be combined with GPR profile [46,47,48].

The most key issue is regarding how the GPR profile converts to the scattered point clouds with high-resolution position information; then, the TLS-based and GPR-derived point clouds are combined into a single dataset for 3D detailed virtual reconstruction. As described in Figure 3, the data fusion method of TLS and GPR mainly includes the following steps. First, registration was used to combine multistation point clouds into a unitary point cloud dataset using the spherical reflectors, which are placed into the overlapping region of the adjacent scanning stations. TLS-based point clouds in the instrument coordinate system were transformed into a geodetic coordinate system by the GCPs, which were acquired using the GNSS static measurement method. The GPR system and GNSS signals receiver were simultaneously triggered to gain the GPR data and geographical coordinates by the pulse signals of the survey wheel. The geographical coordinates, recorded by the DGPS, matched well with each trace of the GPR data using a time synchronization algorithm. Second, the TLS-based point clouds were processed to reduce the noise points and the number of points clouds using the filter and resampling algorithm. In general, the readily identifiable noise points were manually eliminated by Geomagic Studio 12 software, and other unwanted noise points were removed by the point clouds filtering method (the maximum local slope filtering method, Gaussian filtering, Radius Outlier Removal, and other filtering methods). The resampling algorithm was applied to compress the number of point clouds and maintain high-speed computing. Then, the color point clouds were generated by the processed point clouds and the RGB information. In order to reduce the unwanted noises and improve signal-to-noise ratio, the GPR data were processed by subtract-DC-shift, time-zero correction, automatic gain control (AGC), background removal filtering, bandpass filtering, and running average filtering using ReflexW 7.2 software. Time–depth conversion was implemented on each trace of GPR data, and the GPR data were transformed into the scattered point clouds with xyz feature using the improved propagation model of electromagnetic waves. At last, in view of coordinate system transformation, the GPR-derived point clouds were geo-referenced in the same geographic coordinate system as TLS-based point clouds; then, the TLS-based and GPR-derived point clouds (2D GPR data or 3D GPR data) were merged into a single point cloud dataset for 3D detailed virtual reconstruction.

## 3. Methodology

### 3.1. Data Integration Method of GPR and GNSS

In order to obtain high-resolution position information of the GPR data, an integrated GPR and GNSS system was applied to collect the GPR data and their geographical coordinates [49,50,51,52,53]. As shown in Figure 4, this system mainly includes the GPR control unit, the transmitting antenna, the receiving antenna, the survey wheel, the GNSS base station, and the GNSS mobile station. When the GPR antenna was pulled on the surface, the pulse signals of the survey wheel were used to trigger the GPR control unit and the signal receiver of the mobile GNSS station simultaneously. Once a pulse signal reached the GPR control unit, the transmitting antenna emitted an electromagnetic signal into the ground and a single-channel reflection wave was recorded in the GPR system. Meanwhile, the position information of the single-channel reflection wave was recorded in the signal receiver of the GNSS mobile station, such as the geographical coordinates (longitude, latitude, and altitude) and GNSS time. Due to the GPR system and the signal receiver of GNSS being simultaneously triggered by the pulse signals of the survey wheel, the pulse events of the GNSS should be well aligned with the trace of the GPR data. Nevertheless, the trace interval and antenna frequency of the GPR were set before data collection, and it was troublesome that the trace interval was almost perfectly aligned with the calibration of the survey wheel. Therefore, in view of the trace interval and wheel calibration of the survey wheel, a time synchronization algorithm was adopted to achieve data fusion of each trace of the GPR data and the geographical coordinates of the GNSS signal receiver using the GNSS time as a references [45].

Geographical coordinates, recorded by the GNSS signal receiver, were the position information in the geometric center of the GNSS antenna. These geographical coordinates can be expressed by Equation (3).
(3)$${\left[\begin{array}{c} x_{i}\\ y_{i}\\ z_{i} \end{array}\right]}_{GPR}=(1+m)•R(\omega ){\left[\begin{array}{c} x_{i}\\ y_{i}\\ z_{i} \end{array}\right]}_{GNSS}+\left[\begin{array}{c} \Delta x\\ \Delta y\\ \Delta z \end{array}\right]$$
where ${\left(x_{i},y_{i},z_{i}\right)}_{GPR}$ represents the geographical coordinate of a point in the geometric center of the GPR antenna, ${\left(x_{i},y_{i},z_{i}\right)}_{GNSS}$ is the geographic coordinate of a point in the geometric center of the GNSS antenna, R(ω) indicates the rotation matrix between the GPR coordinate system and GNSS coordinate system, *m* is the scale parameter, and $\left(\Delta x,\Delta y,\Delta z\right)$ denotes the translation parameter between the GPR coordinate system and GNSS coordinate system.

As the GNSS antenna is mounted on the top of the GPR antenna, the geometric center of the GPR antenna exhibits good consistency with the geometric center of the GNSS antenna. There is only a translation vector in the vertical direction between the GPR coordinate system and GNSS coordinate system. Consequently, the geographical coordinate in the geometric center of the GPR antenna can be expressed by Equation (4).
(4)$${\left[\begin{array}{c} x_{i}\\ y_{i}\\ z_{i} \end{array}\right]}_{GPR}={\left[\begin{array}{c} x_{i}\\ y_{i}\\ z_{i} \end{array}\right]}_{GNSS}+\left[\begin{array}{c} 0\\ 0\\ \Delta z \end{array}\right]$$
where ${\left(x_{i},y_{i},z_{i}\right)}_{GPR}$ represents the geographical coordinate of a point in the geometric center of the GPR antenna, ${\left(x_{i},y_{i},z_{i}\right)}_{GNSS}$ is the geographic coordinate of a point in the geometric center of the GNSS antenna, and Δz indicates the height value along the Z axes between the geometric center of the GPR antenna and of the GNSS antenna.

### 3.2. The Propagation Model of Electromagnetic Waves

If the average velocity of electromagnetic waves is known, the propagation depth of the reflection waves can be expressed by Equation (5).
(5)$$h=\frac{1}{2}vt$$
where *h* is the propagation depth of electromagnetic waves, *v* implies the average velocity of electromagnetic waves in the media, and *t* represents two-way travel time of electromagnetic waves. This propagation model of electromagnetic waves is relatively simple to depict the propagation characteristics of electromagnetic waves. Owing to the antenna separation between the transmitting antenna and the receiving antenna being unvalued in Equation (5), the true depths of the subsurface materials or buried objects cannot be determined on the GPR profile. This model is commonly used to achieve the detection depth of electromagnetic waves when the antenna separation is adjacent.

For revealing more realistic propagation characteristics of electromagnetic waves, an improved propagation model of electromagnetic waves is illustrated in Figure 5. Compared to Equation (5), the antenna separation and the direct waves in the air were both valued. Considering the polarization characteristics of GPR antenna, the scattering energy of electromagnetic waves can be ignored when the electromagnetic waves are spreading in the media. In this case, the propagation path of electromagnetic waves mainly includes the direct wave in the air and the reflection wave in the ground, paths 1 and 2, respectively, as shown in Figure 5.

The propagation depth of electromagnetic waves (*h*) can be expressed as Equations (6) and (7).
(6)$$h=\sqrt{{\left(\frac{1}{2}vt_{1}\right)}^{2}-{\left(\frac{d}{2}\right)}^{2}}$$
(7)$$t_{1}=t-t_{0}$$
where *v* is the average velocity of electromagnetic waves in the media, h is the propagation depth of the radar reflection wave, $t_{0}$ is two-way travel time of the direct wave in the air, and $t_{1}$ indicates two-way travel time of the radar reflection wave. *d* represents the antenna separation between the transmitting antenna and receiving antenna. Normally, the transmitting antenna and receiving antenna of the GPR are installed together to image the subsurface information with a fixed distance. To optimize the penetration depth and spatial resolution, different frequency GPR antennas often have different antenna separation. After each trace of the GPR image was well aligned with the geographical coordinates, time–depth conversion was realized to obtain the depth position of the sampling points on the GPR data using the improved propagation model of electromagnetic waves.

### 3.3. Coordinate Transformation

Once the GPR data are transformed to the scattered point clouds with high-resolution position information, it is essential to establish the mathematical model between the TLS-based and GPR-derived point clouds for data fusion. Theoretically, the mathematical model can be expressed by Equation (8).
(8)$$\left\|Tp_{i}-q_{i}=0\right\|$$
where $\left(p_{i},q_{i}\right)$ indicates the common points of the TLS-based and GPR-derived point clouds, and matrix *T* represents the transformational matrix between the TLS coordinate system and GPR coordinate system. If the TLS-based point clouds are regarded as the reference dataset, the GPR-derived point clouds can be registered in the geodetic coordinate system of TLS-based point clouds by Equation (9).
(9)$$\left(\begin{array}{c} x_{TLS}\\ y_{TLS}\\ z_{TLS} \end{array}\right)=R(\omega )•S\left(\begin{array}{c} x_{GPR}\\ y_{GPR}\\ z_{GPR} \end{array}\right)+\left(\begin{array}{c} T_{x}\\ T_{y}\\ T_{z} \end{array}\right)$$
where $\left(x_{TLS},y_{TLS},z_{TLS}\right)$ represents the geographical coordinate of a point in the coordinate system of the TLS-based point clouds; $\left(x_{GPR},y_{GPR},z_{GPR}\right)$ indicates the geographical coordinate of a point in the coordinate system of the GPR-derived point clouds; R(ω) represents the rotation matrix between the TLS coordinate system and GPR coordinate system as described in Equation (10); *S* is the scale matrix as shown in Equation (11); and $\left(T_{x},T_{y},T_{z}\right)$ are the translation parameters along the X, Y, and Z axes.
(10)$$R(\omega )=R_{x}•R_{y}•R_{z}=\left[\begin{array}{ccc} \cos\alpha & -\sin\alpha & 0\\ \sin\alpha & \cos\alpha & 0\\ 0 & 0 & 1 \end{array}\right]\left[\begin{array}{ccc} \cos\beta & 0 & \sin\beta \\ 0 & 1 & 0\\ -\sin\beta & 0 & \cos\beta \end{array}\right]\left[\begin{array}{ccc} 1 & 0 & 0\\ 0 & \cos\gamma & -\sin\gamma \\ 0 & \sin\gamma & \cos\gamma \end{array}\right]$$
(11)$$S=\left[\begin{array}{ccc} S_{x} & 0 & 0\\ 0 & S_{y} & 0\\ 0 & 0 & S_{z} \end{array}\right]$$

After the TLS-based and GPR-derived point clouds were registered in the same geometric coordinate system, the GPR data were needed to attribute information normalization for data fusion. In general, surficial information was revealed by the TLS-based point clouds with the position information (x, y, z) and the RGB color information. The (x, y, z) represents the spatial coordinates in the geodetic coordinate system, and the (R, G, B) indicates the color information of point clouds with a value of 0–255 [54]. Nevertheless, the GPR data were generally transformed into the scattered point clouds with the position information (x, y, z) and the attribute information Q. The (x, y, z) represents the spatial coordinates of the sampling points of the GPR image, and the attribute information Q denotes the instantaneous amplitude value of the sampling points, which is determined by the characteristics of subsurface materials or buried objects. The attribute information of the GPR data can be transformed into RGB color information by Equation (12), which is the same as the TLS color point cloud.
(12)$$Q_{0}=255-255\times \frac{Q_{\max}-Q}{Q_{\max}-Q_{\min}}$$
where $Q_{\max}$ indicates the maximum amplitude value of electromagnetic waves on the GPR data, $Q_{\min}$ represents the minimum amplitude value of electromagnetic waves on the GPR data, Q denotes the instantaneous amplitude value of the sampling points, and $Q_{0}$ is the amplitude value of the sampling points after attribute information normalization.

### 3.4. Accuracy Assesment

Considering that the TLS point clouds and GPR data were accurately registered in the same geodetic coordinate system, a comparison of the common points between TLS-based and GPR-derived point clouds was performed to assess the overall accuracy of fusion data. As the common points of the TLS-based and GPR-derived point clouds were almost distributed on the ground, it was essential to extract the ground points from the TLS-based point clouds before assessing the overall accuracy. A progressive triangular irregular network (TIN) densification (PTD) algorithm was selected to extract the ground points and non-ground points from the TLS-based point clouds using the Terrasolid Suite 2019 software [55,56,57].

Once the ground points $P\left(x_{GPR},y_{GPR},z_{GPR}\right)$ of the GPR data were extracted, the corresponding points $P\left(x_{TLS},y_{TLS},z_{TLS}\right)$ can be gained from the ground points of TLS-based point clouds using the iterating calculation by Equation (13).
(13)$$P\left(X_{TLS},Y_{TLS},Z_{TLS}\right)=\min\left(\sqrt{{\left(x-x_{GPR}\right)}^{2}+{\left(y-y_{GPR}\right)}^{2}+{\left(z-z_{GPR}\right)}^{2}}\right)$$
where $\left(x,y,z\right)$ represents the ground points of the TLS-based point clouds.

Finally, the standard deviation (STD) of the commons points between the TLS-based and GPR-derived point clouds were applied to evaluate the overall accuracy of fusion data, as shown in Equation (14).
(14)$$\left\{\begin{array}{c} STD_{x}=\sqrt{\frac{1}{N-1}\sum_{i=1}^{N}{\left(x_{i}-\overset{-}{x}\right)}^{2}}\\ STD_{y}=\sqrt{\frac{1}{N-1}\sum_{i=1}^{N}{\left(y_{i}-\overset{-}{y}\right)}^{2}}\\ STD_{z}=\sqrt{\frac{1}{N-1}\sum_{i=1}^{N}{\left(z_{i}-\overset{-}{z}\right)}^{2}} \end{array}\right.$$
where *N* indicates the number of the commons points between the TLS-based and GPR-derived point clouds; $\left(x_{i},y_{i},z_{i}\right)$ is the 3D coordinate of a common point; and the average value of the commons points along the X, Y, and Z axes can be calculated by Equation (15).
(15)$$\left\{\begin{array}{c} \overset{-}{x}=\frac{1}{N}\left(x_{1}+x_{2}+\ldots +x_{n}\right)\\ \overset{-}{y}=\frac{1}{N}\left(y_{1}+y_{2}+\ldots +y_{n}\right)\\ \overset{-}{z}=\frac{1}{N}\left(z_{1}+z_{2}+\ldots +z_{n}\right) \end{array}\right.$$

## 4. Experimental Results

For evaluating the feasibility and validity of the integrated TLS and GPR method, TLS and GPR (250 MHz and 500 MHz antenna) surveys were implemented to delineate the surface and subsurface geometry of the fault scarp in the Litang fault (30.23° N, 99.86° E). As shown in Figure 6, geomorphic features of the fault scarp were clearly distinguished by topographic change [35,42], and the study site was suitable for evaluating the integration of TLS and GPR to delineate the surficial and subsurface information.

A Faro Focus3D X330 laser scanner was adopted to acquire high-resolution detailed topographic information with a high-integration and easy-operation method (Figure 7a). This phase-shift laser scanner offers massive point clouds with an effective range of 330 m and a ranging error of ±2 mm. With the aim to ensure an unabridged scanning scene, a total of 28 scanning stations were set up with a fixed distance of 30–50 m between the adjacent scanning stations. All point clouds of the scanning stations were combined into a unitary point cloud dataset using Faro Scene 5.2 software with the help of spherical reflectors (Figure 7a), which were placed into the overlapping regions of the adjacent scanning stations. At last, the overall point cloud dataset was transformed in the geodetic coordinate system using the GCPs, which were acquired by the GPS static measurement method.

As described in Figure 7b,c, the subsurface geometry of the fault scarp was obtained using a RAMAC system of MALA Geosciences with a 250 MHz and 500 MHz shielded antenna. In addition, the geographical coordinates of the GPR data were simultaneously gained by the DGPS, which was mounted on the in-between position of the GPR antenna. Four 250 MHz GPR profiles were acquired to depict a general sketch map of shallow geometry associated with the fault scarp (Figure 6 and Figure 8c). Subsequently, ten parallel 500 MHz GPR data were collected to image the 3D shallow geometry of the fault scarp with the same intervals of 1 m, as described in Figure 6 and Figure 8b.

The 3D color point clouds of the fault scarp are shown in Figure 8a. There were many circle blank areas on the point clouds, indicating the TLS scanning stations. The 3D GPR data were generated using ten parallel 500 MHz GPR profiles with the same intervals of 1 m, as shown in Figure 8b (yellow rectangle in Figure 6). After the GPR data were transformed into the scattered point clouds with high-resolution position information, the TLS point clouds and GPR data (2D and 3D) were merged together by the data integration method of TLS and GPR. Data visualizations of TLS-based and GPR-derived point clouds (2D and 3D) are shown in Figure 8c,d. The realistic surface and subsurface geometry of the fault scarp were depicted using the integration data of TLS and GPR. Furthermore, the subsurface information of the fault scarp can be better interpreted and understood with the help of its corresponding geomorphic features.

The common points of the GPR data were chosen from four 500 MHz 2D GPR data with the same intervals of 1 m, and 10 common points were extracted from each GPR profile along the measurement lines with the same interval. After the common points of the GPR data are confirmed, the corresponding points of the TLS-based point clouds can be obtained using Equation (13). As shown in Figure 9, a total of 40 common points between the TLS-based and GPR-derived point clouds were implemented to assess the overall accuracy of fusion data. In addition, the $STD_{x}$, $STD_{y}$, and $STD_{z}$ of the common points between the TLS-based and GPR-derived point clouds were 0.9 cm, 0.8 cm, and 2.9 cm, respectively. From the difference values in Figure 9, the maximum difference values in the x, y, and z directions were no more than 5 cm. The difference values in the x and y directions were relatively stable within 2 cm, while the difference values in the z direction had an abrupt fluctuation and the maximum values could be up to 5 cm. In view of the difference values and the STD in the x, y, and z directions, field experimental results demonstrate that the GPR-derived point clouds exhibit good consistency with the TLS-based point clouds, and the feasibility and overall accuracy of the integrated TLS and GPR method were also well verified.

## 5. Discussion

Generally, it is difficult to obtain detailed surface and subsurface information for 3D detailed virtual reconstruction using conventional surveying methods. Advanced technologies such as TLS and GPR have been increasingly used to acquire high-resolution geomorphic features and shallow geometry in many fields. TLS is capable of providing various levels of point clouds to delineate real-world objects or environments with a highly automated and non-contacted operation. GPR is a nondestructive geophysical technique to image the shallower buried structures using high-frequency electromagnetic waves. However, the TLS and GPR methods have their own advantages and drawbacks. It is clear that the shallow geometry of subsurface materials cannot be obtained by the TLS without an intrusive intervention. In addition, the understanding of GPR data is unintelligible considerable the remarkable variation in radar waveforms and relative amplitude. Therefore, it is worth paying great attention to implementing data integration of TLS and GPR for comprehensive identification and analysis. Compared with single TLS or GPR data, the advantages of fusion data of TLS and GPR are summarized as follows: (1) it offers multisensor and multiscale spatial data for the comprehensive identification and analysis of surficial and subsurface information; (2) shallower buried structures obtained by the GPR will be better interpretated and understood with the help of their corresponding superficial spatial data.

The proposed methodology of this study offers a reliable approach to integrate TLS and GPR data for 3D detailed virtual reconstruction. Compared with other data fusion methods, the GPR data with high-resolution position information at the centimeter level were transformed into dense point clouds rather than the 2D image. The GPR data with high-resolution position information were simultaneously gathered using the pulse signals of the survey wheel fixed on the GPR antenna. In addition, a time synchronization algorithm was used to combine each trace of the GPR data with its position information acquired by the DGPS. Based on the improved propagation model of electromagnetic waves, the TLS-based and GPR-derived point clouds were merged into a single point cloud dataset using coordinate transformation. Furthermore, TLS and GPR (250 MHz and 500 MHz antenna) surveys were conducted to assess the feasibility and overall accuracy of the proposed methodology. In view of the difference values and the STD in the x, y, and z directions, field experimental results demonstrate that the GPR-derived point clouds exhibit good consistency with the TLS-based point clouds.

Although the overall accuracy of the proposed methodology was evaluated by the common points between the TLS-based and GPR-derived point clouds, a comparison between the fusion data and reference measurements (such as the total station, RTK-GPS and trench excavation, etc.) was not performed because of the severe natural environment with high altitudes and very few CORS reference stations nearby the study site. Moreover, the GPR had a greater investigation depth than trench excavation, and shallower buried structures obtained by GPR cannot be fully verified by the traditional method, especially on the Qinghai–Tibet Plateau with its fragile ecological environment. Data fusion of TLS and GPR was initially realized by the scattered point clouds in this research; the comparison between the integrated data and the reference measurements will be the subject of deeper analysis in a further study. Some errors were present on the TLS data post-processing procedures. On the one hand, when multistation point clouds are combined into a unitary point cloud dataset using the spherical reflectors on the ground, the folding error at a centimeter level may be generated in the TLS point clouds and will increase as the number of the scanning stations increases. On the other hand, due to there being few CORS reference stations near the study site, the overall point cloud dataset was transformed in the geodetic coordinate system with the help of the GCPs using the GNSS static measurement method. The georeferencing error may also be produced in the TLS point clouds. Furthermore, there were also some errors present on the GPR data post-processing procedures, especially in the propagation model of electromagnetic waves. When the terrain is relatively flat, the improved propagation model of electromagnetic waves in this study can provide good results for data integration of TLS and GPR. Once the terrain was fluctuant, the tilt angle of the GPR antenna had an effect on the accurate positioning of the subsurface materials or buried objects. As a result, the overall accuracy of the proposed methodology was determined by many factors such as the surveying conditions, data acquisition procedures, data post-processing procedures, and project significance.

The study results demonstrate that the proposed methodology in data fusion of TLS and GPR is capable of providing detailed surface and subsurface information for 3D detailed virtual reconstruction. Nevertheless, though data integration of TLS and GPR was initially realized by the scattered point clouds, the data integration of GPR data with other spatial data—such as DEM, remote sensing image, airborne LiDAR, and so on—is still needed for a deeper analysis.

## 6. Conclusions

In this work, a reliable systematic methodology of data integration of TLS and GPR was proposed for 3D detailed virtual reconstruction. With the help of the pulse signals of the survey wheel, the GPR profiles and high-precision geographical coordinates were simultaneously gathered using the GPR system and the GNSS signal receiver. Based on the characteristic of TLS point clouds and GPR data, the GPR data with high-resolution position information were transformed into dense point clouds using the propagation model of electromagnetic waves. The TLS-based and GPR-derived point clouds were merged into a single point cloud dataset using coordinate transformation. The methodology we introduced exhibits more accurate and reliable performance for data integration of TLS and GPR. TLS and GPR (250 MHz and 500 MHz antenna) surveys were conducted in the Litang fault to assess the feasibility and overall accuracy of the proposed methodology. This method also offers the chance for data integration of GPR data with other spatial data, such as the Digital Elevation Model (DEM), unmanned aerial vehicle photogrammetry, airborne LiDAR, etc.

## Figures and Tables

**Figure 2 sensors-23-09826-f002:**
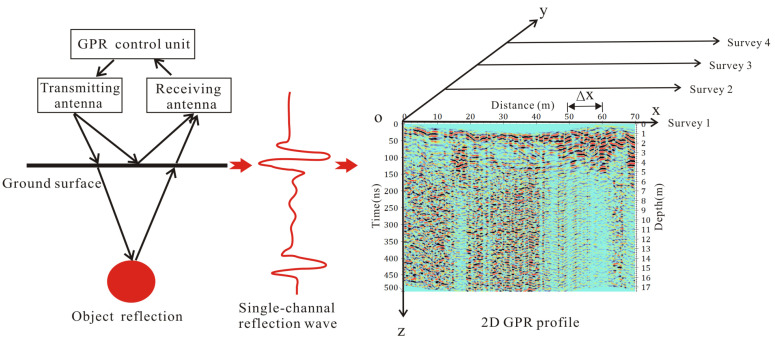
Working principle of the GPR and the spatial coordinate system of GPR data [45]. The x axis represents the horizontal distance of GPR profile, the y axis shows the number of GPR profiles along different survey lines, and the z axis is the two-way travel time of electromagnetic waves. Red color on the 2D GPR profile indicates high-amplitude radar reflections; green color represents low-amplitude radar reflections.

**Figure 3 sensors-23-09826-f003:**
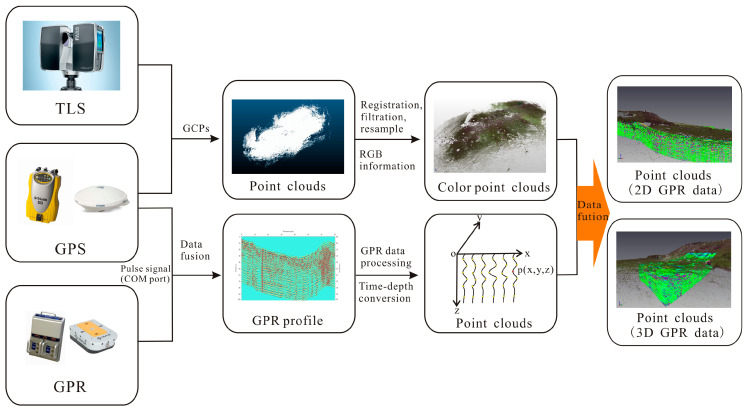
Data integration workflow of TLS and GPR. Dark color (red or blue) on the GPR data indicates high-amplitude radar reflections, and light color (green) represents low-amplitude radar reflections.

**Figure 4 sensors-23-09826-f004:**
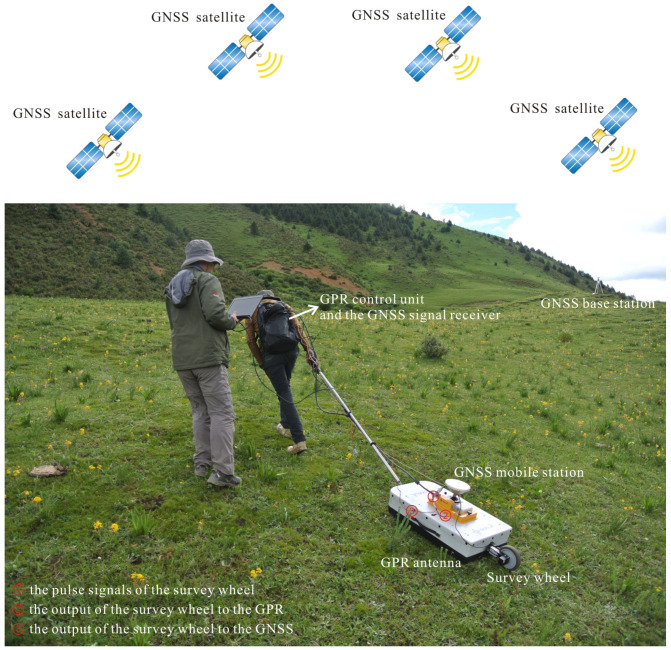
Data acquisition system of GPR and DGPS [45]. The GNSS antenna was situated on the in-between position of the GPR antenna. The pulse signals of the survey wheel were used to trigger the GPR control unit and the signal receiver of the mobile GNSS station. The GPR data and their geographical coordinates were simultaneously acquired using the GPR system and the signal receiver of the mobile GNSS station.

**Figure 5 sensors-23-09826-f005:**
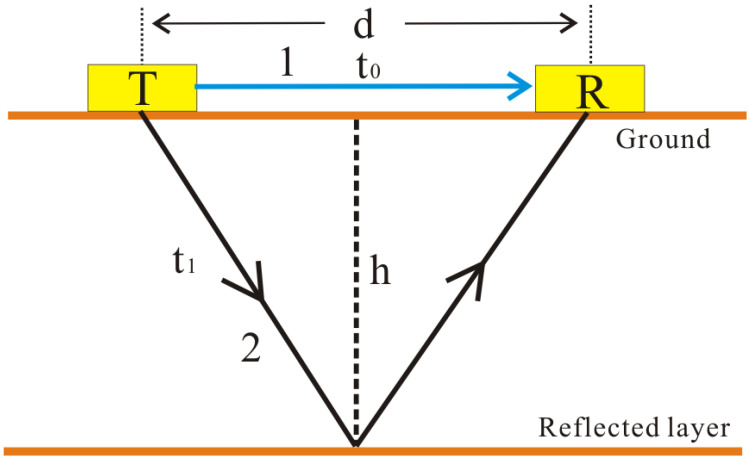
The improved propagation model of electromagnetic waves. Path 1 indicates the direct wave in the air and path 2 represents the reflection wave in the ground.

**Figure 6 sensors-23-09826-f006:**
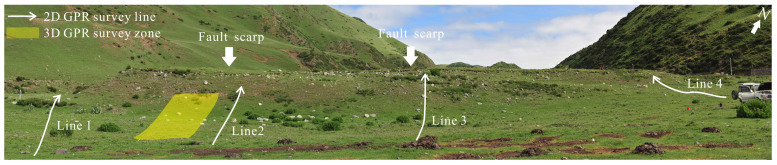
Panoramic image of the study site. White lines indicate four 2D 250 MHz GPR survey lines, and yellow rectangle represents the 3D GPR survey zone.

**Figure 7 sensors-23-09826-f007:**
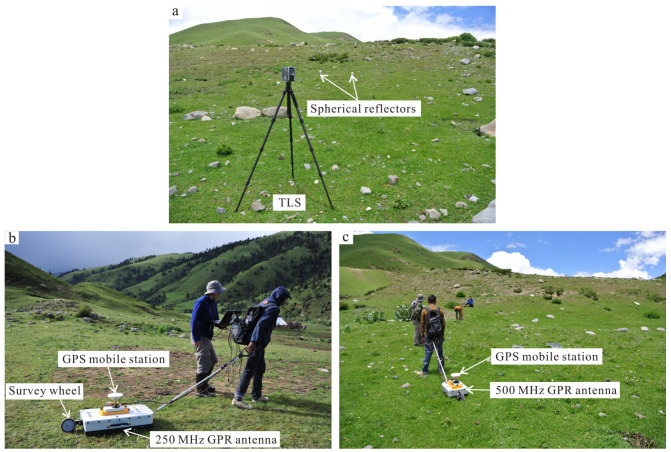
TLS and GPR data acquisition. (**a**) TLS data acquisition. (**b**) GPR data acquisition with 250 MHz shield antenna. (**c**) GPR data acquisition with 500 MHz shield antenna.

**Figure 8 sensors-23-09826-f008:**
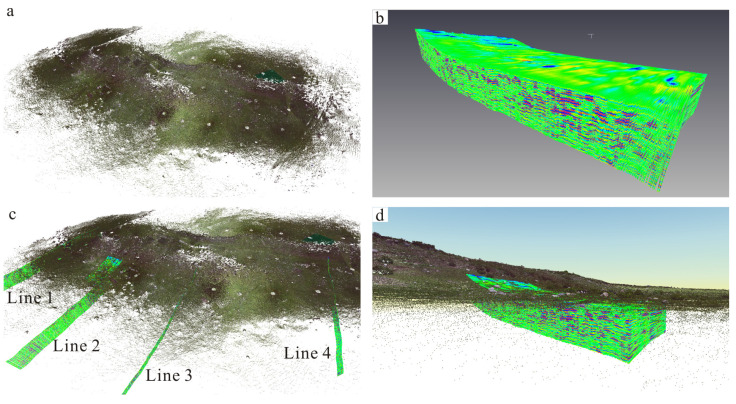
Data integration of TLS and GPR. Blue color on the GPR data indicates high-amplitude radar reflections, and green color represents low-amplitude radar reflections. (**a**) TLS-based point clouds. (**b**) 3D GPR data. The 3D data were generated by ten parallel 500 MHz GPR profiles with the same intervals of 1 m. (**c**) Data visualization of TLS-based and GPR-derived point clouds (2D and 3D). (**d**) Data visualization of TLS point clouds and 3D GPR data.

**Figure 9 sensors-23-09826-f009:**
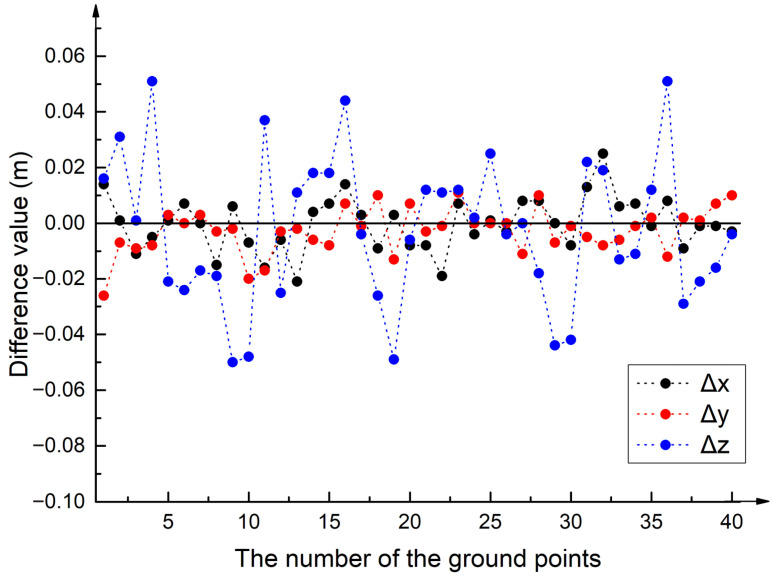
Difference values of the common points between the TLS-based and GPR-derived point clouds. Black dots represent the difference values in the x direction, red dots indicate the difference values in the y direction, and blue dots represent the difference values in the z direction.

## Data Availability

The data presented in this study are available on request from the corresponding author. The data are not publicly available due to privacy.

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
