# Peer review of "Multisensor and Multiscale Data Integration Method of TLS and GPR for Three-Dimensional Detailed Virtual Reconstruction"

_sensors, 2023, doi:10.3390/s23249826_

Round 1

Reviewer 1 Report

Comments and Suggestions for Authors

1. Abstract should contain more information from the field of results, concrete values, accuracy, etc., so as to convince to delve into the content of the publication - without engineering arguments, i.e. values, it is hardly convincing

2. please divide the literature review into application areas and summarise each one in a sentence with conclusions/persuasions, rather than collectively giving refs. 1-6, without going into details, same for line 41

3. 46 references in a manuscript of 16 pages is not an overwhelming number - I suggest using the literature on the topics:

GPR:

https://scholar.google.com/citations?user=BJBjnPYAAAAJ&hl=pl&oi=ao

https://www.sciencedirect.com/science/article/pii/S0263224122008570

data integration:

http://ptfit.sgp.geodezja.org.pl/wydawnictwa/kazimierz-2013/22_Warchol_255_260.pdf

4. line 78 - " high-resolution position information" - instead of a general expression please give at least an order of magnitude - 1cm, 1 dm or 1m

5. line 84 - is this really a 3D reconstruction model? a point cloud and a model are two completely different entities! same in line 181, the same 451, and 466 - model allows us to specify a value at any point (like a function), and with point clouds we have discrete representation of space

6. I would not use the expression "texture" in the context of colouring point clouds from photos, because "texture" suggests continuity of surface, whereas here we have information about RGB values assigned to individual points, so further on LiDAR point clouds are a discrete representation of space and not a model, even less a textured one, the same 154 line - "and the RGB colour information" and 301 and 310

7. line 114 - convert is not the right word here, as this refers more to data formats, please use rather "transformed" or "recalculated"

8. please consider where in the manuscript the acronym GNSS should be instead of GPS - in the part describing the experiment - GPS - OK, but in the others it should be GNSS, because in the method there is no restriction on the issue of global coordinate providers

9. fig. 4. - at the top there should be not 3 satellites but 4, because so many are needed to determine the position (including clock synchronisation) - 3 satellites are misleading

10. maybe it would be worthwhile to consider other ways of classifying ground class for TLS, not just based on slope - e.g. Axelsson algorithm

11. line 361 - why is the precision given as +- 5mm at 50m, if according to the manufacturer the distance measurement error is +- 2mm, if the value is given based on research then please provide reference

12. line 367 - what was the folding error of the TLS design in Faro Scene?

13. line 368 - how many GCPs were used for georeferencing? what was the georeferencing error for TLS

14. what was the GPR-derived point cloud density? and/or avg distance between points?

Author Response

Thank you very much for your letter and the advices about our manuscript entitled “Multi-sensor and Multi-scale Data Integration Method of TLS and GPR for Reconstructing the Three-dimensional Realistic Model” (sensors-2691917) submitted to Sensor. These comments and advices are very valuable to improve our manuscript. After carefully studying the comments, we have made the revisions by the reviewers' comments.

Point 1: Abstract should contain more information from the field of results, concrete values, accuracy, etc., so as to convince to delve into the content of the publication - without engineering arguments, i.e. values, it is hardly convincing.

Response 1: The context “A total of 40 common points between the TLS-based point cloud and GPR-derived point cloud were implemented to assess the data fusion accuracy. The difference values in x, y directions were relatively stable within 2 cm, while the difference values in z direction had an abrupt fluctuation and the maximum values could up to 5 cm. The Standard Deviation (STD) of the common points between the TLS-based point cloud and GPR-derived point cloud were 0.9 cm, 0.8 cm and 2.9 cm, respectively.” has been added in the Abstract.

Point 2:  please divide the literature review into application areas and summarise each one in a sentence with conclusions/persuasions, rather than collectively giving refs. 1-6, without going into details, same for line 41.

Response 2: The literature review has been divided into application areas and summarise each one in a sentence in line 37 and 41.

The context in line 37 (refs. 1-6)” In contrast with traditional methods, TLS is generally used to acquire the detailed spatial information for reconstructing 3D realistic model because it allows offering high-resolution dense point cloud with highly automated and non-contacted operation in a short time” has been revised as “In contrast with traditional methods, TLS is generally used to acquire the detailed spatial information for reconstructing 3D realistic model because it allows offering high-resolution dense point cloud with highly automated and non-contacted operation in a short time, such as in the geosciences [1-4], architecture [5-7], archaeology [8, 9], agriculture and forestry [10], engineering and construction industry [11, 12] and so on.”.

The context in line 41 ” GPR has been commonly chosen to delineate the shallow geometry of subsurface materials or buried-object with a non-destructive and cost-effective fashion ” has been revised as “GPR has been commonly chosen to delineate the shallow geometry of subsurface materials or buried-object with a non-destructive and cost-effective fashion. This technology has been widely applied in the geology investigation [13-16], cultural heritage protection [9, 17-19], urban infrastructure detection [20-23], water dynamics investigation [24] and so on.”.

Point 3: 46 references in a manuscript of 16 pages is not an overwhelming number - I suggest using the literature on the topics:

GPR:

https://scholar.google.com/citations?user=BJBjnPYAAAAJ&hl=pl&oi=ao

https://www.sciencedirect.com/science/article/pii/S0263224122008570

data integration:

http://ptfit.sgp.geodezja.org.pl/wydawnictwa/kazimierz-2013/22_Warchol_255_260.pdf

Response 3: Four references has been added in the revised manuscript as following:

  1. Gabryś, M.; Ortyl, Ł., Georeferencing of Multi-Channel GPR—Accuracy and Efficiency of Mapping of Underground Utility Networks. Remote Sens 2020, 12 (18), 2945.
  2. Marta, G.; Katarzyna, K.; Ortyl, Ł., GPR surveying method as a tool for geodetic verification of GESUT database of utilities in the light of BSI PAS128. Reports on Geodesy and Geoinformatics 2019, 107, 49-59.
  3. Janos, D.; Kuras, P.; Ortyl, Ł., Evaluation of low-cost RTK GNSS receiver in motion under demanding conditions. Measurement 2022, 201, 111647.
  4. Warchoł, A., Analysis of accuracy airborne, terrestrial and mobile laser scanning data as an introduction to their integration. Archiwum Fotogrametrii, Kartografii i Teledetekcji 2013, 25, 255-260.

Point 4: line 78 - " high-resolution position information" - instead of a general expression please give at least an order of magnitude - 1cm, 1 dm or 1m.

Response 4: The context in line 78 “The GPR data with high-resolution position information were simultaneously gathered by the pulse signals of the survey wheel.” has been revised “The GPR data with high-resolution position information on the centimeter level were simultaneously gathered by the pulse signals of the survey wheel.”

Point 5: line 84 - is this really a 3D reconstruction model? a point cloud and a model are two completely different entities! same in line 181, the same 451, and 466 - model allows us to specify a value at any point (like a function), and with point clouds we have discrete representation of space.

Response 5:  The context in line 84 “the TLS-based point cloud and GPR-derived point cloud were merged into a single point cloud dataset for data visualization and 3D model reconstruction”has been rivised “the TLS-based point cloud and GPR-derived point cloud were merged into a single point cloud dataset for 3D detailed virtual reconstruction”.

The context in line 181 “and the TLS-based point cloud and GPR-derived point cloud (2D GPR profile or 3D GPR data) were merged into a single point cloud dataset for 3D visualization and model reconstruction.”has been rivised “the TLS-based point cloud and GPR-derived point cloud were merged into a single point cloud dataset for 3D detailed virtual reconstruction”.

The context in line 451” The study results demonstrate that the proposed methodology in the data integration of TLS and GPR has the capable of providing detailed surface and subsurface information for reconstructing 3D realistic model”has been rivised “The study results demonstrate that the proposed methodology in the data integration of TLS and GPR has the capable of providing detailed surface and subsurface information for 3D detailed virtual reconstruction”.

The context in line 466” In this work, a reliable systematic methodology of data integration of TLS and GPR was proposed for reconstructing the detailed 3D realistic model”has been rivised “In this work, a reliable systematic methodology of data integration of TLS and GPR was proposed for 3D detailed virtual reconstruction”.

Point 6: I would not use the expression "texture" in the context of coloring point clouds from photos, because "texture" suggests continuity of surface, whereas here we have information about RGB values assigned to individual points, so further on LiDAR point clouds are a discrete representation of space and not a model, even less a textured one, the same 154 line - "and the RGB colour information" and 301 and 310.

Response 6: The context “the color texture information” in line 154 has been revised as “the RGB value”.

The context in “texture information (R, G, B).” line 301 and 310 has been revised as “the RGB colour information”.

Point 7: line 114 - convert is not the right word here, as this refers more to data formats, please use rather "transformed" or "recalculated"

Response 7: The context “In addition, the point cloud in the instrument coordinate system can be converted into the geodetic coordinate system by ground control points (GCPs)” has been revised “In addition, the point cloud in the instrument coordinate system can be transformed into the geodetic coordinate system by ground control points (GCPs)”.

Point 8: please consider where in the manuscript the acronym GNSS should be instead of GPS - in the part describing the experiment - GPS - OK, but in the others it should be GNSS, because in the method there is no restriction on the issue of global coordinate providers.

Response 8: The GNSS has been used to be instead of GPS except on the part describing the experimen. In addtion, the GPS has been revised as GNSS in Fig.3 and Fig. 4.

Point 9:  fig. 4. - at the top there should be not 3 satellites but 4, because so many are needed to determine the position (including clock synchronisation) - 3 satellites are misleading.

Response 9: One satellite has been added in the Fig 4 for determining the position as shown in revised manuscript.

Point 10:  maybe it would be worthwhile to consider other ways of classifying ground class for TLS, not just based on slope - e.g. Axelsson algorithm.

Response 10:  A progressive triangular irregular networks (TIN) densification (PTD) algorithm was selected to extract the ground points and non-ground points from the TLS-based point cloud using the Terrasolid software.

Point 11:  line 361 - why is the precision given as +- 5mm at 50m, if according to the manufacturer the distance measurement error is +- 2mm, if the value is given based on research then please provide reference.

Response 11: On the basis of the User Manuals of Focus3D X330 and X330HDR, the ranging error has been revised as ±2 mm in the revised manuscript.

(https://knowledge.faro.com/Hardware/Focus/Focus/User_Manuals_and_Quick_Start_Guides_for_the_Focus_Laser_Scanner?mt-learningpath=focus_m70_s150-350_downloads)

The context “This phase-shift laser scanner allows collecting a massive point cloud with an effective range of 330 m and a precision of ±5 mm at a range of 50 m” has been revised as” This phase-shift laser scanner allows collecting a massive point cloud with an effective range of 330 m and a ranging error of ±2 mm” in line 361.

Point 12:  line 367 - what was the folding error of the TLS design in Faro Scene?

Response 12: A total of 28 scanning stations were performed with a fixed distance of 30-50m between the adjacent scanning stations. All point cloud of the scanning stations were combined into a unitary point cloud dataset using the Faro Scene software. The folding error of the TLS design in Faro Scene was about ± 1cm.

Point 13: line 368 - how many GCPs were used for georeferencing? What was the georeferencing error for TLS

Response 13: Four GCPs were used for georeferencing. Because the suvey site in the Qinghai-Tibet Plateau, there were no CORS reference station nearby the study site. The geodetic coordinates of fourGCPs were obtained by the GPS static measurement method using the Inertial Explorer 8.4 software. The georeferencing error for TLS was up to centimetre-level.

Point 14:  what was the GPR-derived point cloud density? and/or avg distance between points?

Response 14: Ten parallel 500 MHz GPR data were collected with the same intervals of 1 m for depicting the 3D shallow geometry with 0.05 m trace interval. The vertical resolution of the sampling points on the 500 MHz GPR data was about 0.05m. The GPR-derived point cloud density was shown in the following Fig.

Reviewer 2 Report

Comments and Suggestions for Authors

The paper highlights the need for a systematic method to integrate terrestrial laser scanning (TLS) and ground-penetrating radar (GPR) data. It presents a technique for creating detailed 3D models by synchronizing pulse signals from a survey wheel to gather GPR profiles and precise geographic coordinates. The study successfully demonstrates the feasibility and accuracy of this integrated TLS and GPR approach for detailed 3D modeling. Based on the authors' attainment of satisfactory results and the potential usefulness of the study to researchers, I recommend the publication of this study in Sensors, with the inclusion of the following comments.

1.      The abstract falls short of providing a comprehensive overview of the study. It is essential to encompass the study's importance, the research issue addressed, the employed methodology, key findings, and potential future implications of the results.

2.      The introduction should be extended by adding more relevant studies or discussing the existing literature study not just adding the reference, by highlighting the research need, research gap, and novelty, and clarifying the significance of the presented study, which is currently not evident.

3.      How does the simultaneous collection of GPR profiles and geographical coordinates contribute to the creation of a detailed 3D model?

4.      Elaborate on the procedures involved in the data processing and filtering techniques applied to the TLS and GPR data.

5.      How did you validate the accuracy and reliability of the integrated TLS and GPR data?

6.      The study lacks a thorough discussion of the obtained results; the discussion part should be extended by describing the implementation of the current study.

7.      Can you provide details on the ground truth data or reference measurements used for comparison with the integrated data?

8.      Describe the potential applications or future research directions for the integrated TLS and GPR method.

Comments on the Quality of English Language

The English language proficiency is acceptable and meets the required standards.

Author Response

Many thanks for the insightful comments and suggestions about our manuscript entitled “Multi-sensor and Multi-scale Data Integration Method of TLS and GPR for Reconstructing the Three-dimensional Realistic Model” (sensors-2691917) submitted to Sensor. We tried our best to improve our manuscript in the revised manuscript.

Point 1:  The abstract falls short of providing a comprehensive overview of the study. It is essential to encompass the study's importance, the research issue addressed, the employed methodology, key findings, and potential future implications of the results.

Response 1: The abstract has been revised as “The integrated TLS and GPR data could provide the multi-sensor and multi-scale spatial data for comprehensive identification and analyzation of surficial and subsurface information, but a reliable systematic methodology associated to data integration of TLS and GPR was still scarce. The aim of this research was to develop a methodology in data integration of TLS and GPR for three-dimensional (3D) detailed virtual reconstruction. The GPR data and high-precision geo-graphical coordinates on the centimeter level were simultaneously gathered by the GPR system and the GNSS (Global Navigation Satellite System) signal receiver. Time synchronization algo-rithm was proposed to combine each trace of GPR data with its position information. In view of the improved propagation model of electromagnetic wave, the GPR data was transformed into the dense point cloud in the geodetic coordinate system. Finally, the TLS-based point cloud and GPR-derived point cloud were merged into a single point cloud dataset by coordinate transfor-mation. In addition, the TLS and GPR (250 MHz and 500 MHz antenna) surveys were conducted in the Litang fault to assess the feasibility and overall accuracy of the proposed methodology. The 3D realistic surface and subsurface geometry of fault scarp were displayed by the integration data of TLS and GPR. A total of 40 common points between the TLS-based point cloud and GPR-derived point cloud were implemented to assess the data fusion accuracy. The difference values in x, y di-rections were relatively stable within 2 cm, while the difference values in z direction had an abrupt fluctuation and the maximum values could up to 5 cm. The Standard Deviation (STD) of the common points between the TLS-based point cloud and GPR-derived point cloud were 0.9 cm, 0.8 cm and 2.9 cm, respectively. Based on the difference values and the Standard Deviation (STD) in x, y, z directions, the field experimental results demonstrate that the GPR-derived point cloud exhibits well consistency with the TLS-based point cloud. What’s more, this study also offers a good future prospect of the integration method of TLS and GPR for comprehensive interpreting and analyzing the surficial and subsurface information in many fields, such as archaeology, urban infrastructure detection, geological investigation and other fields.” in the revised manuscript.

Point 2:  The introduction should be extended by adding more relevant studies or discussing the existing literature study not just adding the reference, by highlighting the research need, research gap, and novelty, and clarifying the significance of the presented study, which is currently not evident.

Response 2: More relevant studies and discussing the existing literature study has been added in the introduction. The context “Spahic et al. [39] rebulided the 3D model of fault outcrop in the Gocad software by the T-LiDAR point cloud and GPR images. Bubek et al. [35] combined the TLS topography and 2D GPR image to determine the boundary of the hanging wall and the footwall of normal fault. Dlesk. [40] intergrated TLS and GPR data in Rhinocereos 3D software for 3D documentation of historical tunnels. Ercoli et al. [41] combined the TLS point cloud and 2D GPR images for investigating ancient polygonal walls of Amelia. Cowie et al. [42] constrained the loation of the fault plane based on the TLS and GPR data for obtaining the vertical displacement of the normal fault. Puente et al. [9] present the 3D reconstruction of the Roman Site “Aquis Querquennis” based on the fusion data of GPR, T-LiDAR and IRT, and the T-LiDAR orthoimages were combined with GPR time-slices for comprehensive interpretation. Zhang et al. [43] provided the data visualization of fault scarps in Bentley Pointools software using the TLS point clouds and GPR data. Zhang et al. [37] used the TLS-derived data and 2D GPR images for reconstrucing the 3D suficial and subsurface model of Yushu fault. Previous research on data fusion method of TLS and GPR mainly focuses on two aspects. First, the 2D GPR images were directly overlay on the TLS-based point cloud by computer vision theory [35, 40, 41, 44]. It has an advantage of quickly realizing the data integration of TLS and GPR with different data formats, but the multi-perspective spatial data was only displayed in the same scene. Second, the GPR data was transformed to the scatted point cloud, and then the point cloud and GPR data were merged into a single point cloud dataset for data visualization [9, 37, 39, 43]. Although a limited number of research work have present some cases for data integration of TLS and GPR in the scatted point cloud form, the reliable systematic methodology was still scarce. Most of these integrated method were initially realized in the scatted point cloud form by the ground control points, the accuracy of data fusion cannot be ensured. What’s more, the GPR-derived point cloud with high position information was essential for data fusion of TLS and GPR. Considering the GPR data acquirmentition and formats, it is difficult to combine each trace of the GPR profile with its position information acquired by DGPS.” has been added in the Introduction.

Point 3:  How does the simultaneous collection of GPR profiles and geographical coordinates contribute to the creation of a detailed 3D model?

Response 3: The simultaneous collection of GPR profiles and geographical coordinates is crucial to obtain high-resolution position information of GPR data. Once each trace on the GPR image was in well matching with the the geographical coordinates of the GNSS signal receiver, the data fusion of TLS-based point cloud and GPR data can be achived by the coordinate system transformation,and the GPR-derived point cloud is geo-referenced in the same geographic coordinate system as TLS-based point cloud for 3D virtual visualization.

In this study, the integrated GPR and GNSS system were applied to collect the GPR data with its geographical coordinates at the same time. When the GPR is pulled on the ground surface, the pulse signals of the survey wheel are applied to trigger the GPR control unit and the signal communication ports of the GPS signal receiver at the same time. Due to the GPR and the signal receiver of GNSS were simultaneously controlled by the pulse signals of the survey wheel, it is theoretically possible that the pulse events of the GNSS should be well aligned with the number of the trace on the GPR data.

Point 4: Elaborate on the procedures involved in the data processing and filtering techniques applied to the TLS and GPR data.

Response 4:  The context “First, registration was conducted to combine multi-stations scanning point cloud into a unitary point cloud dataset by the spherical reflectors, which placed into the overlapping region of the adjacent scanning stations. “ and “Second, the TLS-based point cloud was processed by the filter and resampling algorithm to reduce the noise points and the number of points cloud. In general, the readily identifiable noise points were manually eliminated by Geomagic Studio software, and other unwanted noise points were removed by the point cloud filtering (The maximum local slope filtering method, Gaussian filtering, RadiusOutlierRemoval and other filtering method). The resampling algorithm was appied to compress the number of points cloud and maintain high speed computing.”has been added in in section “2.3. Data integrated principle of TLS and GPR” in the revised manuscript.

The context ” Subsequently, in order to reduce the unwanted noise and improve the signal-to-noise ratio, the GPR data were processed by subtract-DC-shift, time-zero correction, automatic gain control (AGC), background removal filtering, bandpass filtering and running average filtering using the ReflexW 7.2 software.”has been added in section “2.3. Data integrated principle of TLS and GPR” in the revised manuscript.

Point 5: How did you validate the accuracy and reliability of the integrated TLS and GPR data?

Response 5: Considering the TLS point cloud and GPR data were accurately registered in the same geodetic coordinate system, the comparison of the common points between TLS-based point cloud and GPR-derived point cloud was performed to assess the accuracy of the integrated TLS and GPR method.In the study site, a total of 40 common points between the TLS-based point cloud and GPR-derived point cloud were implemented to assess the data fusion accuracy. The difference values in x, y directions were relatively stable within 2 cm, while the difference values in z direction had an abrupt fluctuation and the maximum values could up to 5 cm. In addition, the Standard Deviation (STD) of the common points between the TLS-based point cloud and GPR-derived point cloud were 0.9 cm, 0.8 cm and 2.9 cm, respectively. Based on the difference values in x, y ,z directions and the Standard Deviation (STD), the field experimental results demonstrate that the GPR-derived point cloud exhibits well consistency with the TLS-based point cloud, and the feasi-bility and accuracy of the proposed methodology in the data integration of TLS and GPR were also well verified.

Point 6: The study lacks a thorough discussion of the obtained results; the discussion part should be extended by describing the implementation of the current study.

Response 6: The context “Although the overall accuracy of the proposed methodology was evaluated by the common points between the TLS-based point cloud and GPR-derived point cloud, the comparison between the fusion data and the reference measurments (such as the total station, RTK-GPS and trench excavation and so on) was not perfomed in the study site because of the severe natural environment with high altitudes and very few CORS reference station nearby the study site. Moreover, the GPR had a more investigation depth than the trench excavation, and the shallower buried structures obtained by GPR can not be fully verified by the traditional method, especailly on the Qinghai-Tibet Plateau with the fragile ecological environment. Data integration of TLS and GPR was initially realized by the scattered point cloud in this research, the comparison between the integrated data and the reference measurments will be deeper analysis in the further study. There were some errors exsited on the TLS data post-processing procedures. On the one hand, when the multi-station point cloud were combined into a unitary point cloud dataset by the spherical reflectors on the ground, the folding error with a centimeter level may be generated in the TLS point cloud and it will be increased as the number of the scanning stations. On the other hand, due to few CORS reference station nearby the study site, the overall point cloud dataset were transformed in the geodetic coordinate system with the help of the ground control points (GCPs) acquired by the GPS static measurement method. The georeferencing error may be also produced in the TLS point cloud. What’s more, there were also some errors exsited on the GPR data post-processing procedures, especially the propagation model of electromagnetic wave. When the terrain is relatively flat, the improved propagation model of electromagnetic wave in this study could provides a good results for data integration of TLS and GPR.  Onec the terrain was fluctuant along the GPR survey lines, the tile angle of GPR antenna had an effect on the accurate positioning of the subsurface materials or buried-object on the GPR image. As a result, the overal accuracy of the proposed methodology was determined by many factors such as the surveying conditons, data acquisition procedures, data post-processing procedures and project significance. ” has been added in the section “5. Discussion”.

Point 7:  Can you provide details on the ground truth data or reference measurements used for comparison with the integrated data?

Response 7: The 40 common points between the TLS-based point cloud and GPR-derived point cloud were provided in the following table. The difference value in x, y,z direction between the TLS-based point cloud and GPR-derived point cloud can be caculted by the spatial coordinates of common points. The Standard Deviation (STD) of the common points between the TLS-based point cloud and GPR-derived point cloud were 0.9 cm, 0.8 cm and 2.9 cm, respectively.

 Table 1  The difference value of 40 common points between the TLS-based point cloud and GPR-derived point cloud

Point number

  x

      TLS

y

z

x

GPR

y

z

差值

△x

△y

△z

1.  582635.587

3346006.241

4158.624

582635.573

3346006.267

4158.608

0.014

-0.026

0.016

2.  582634.418

3346008.21

4158.839

582634.417

3346008.217

4158.808

0.001

-0.007

0.031

3.  582633.254

3346009.825

4158.936

582633.265

3346009.834

4158.935

-0.011

-0.009

0.001

4.  582632.302

3346011.404

4159.121

582632.307

3346011.412

4159.07

-0.005

-0.008

0.051

5.  582631.237

3346013.204

4159.26

582631.236

3346013.201

4159.281

0.001

0.003

-0.021

6.  582629.912

3346014.983

4159.476

582629.905

3346014.983

4159.5

0.007

0

-0.024

7.  582628.728

3346016.606

4159.837

582628.728

3346016.603

4159.854

0

0.003

-0.017

8.  582627.669

3346018.206

4160.397

582627.684

3346018.209

4160.416

-0.015

-0.003

-0.019

9.  582626.883

3346019.299

4160.847

582626.877

3346019.301

4160.897

0.006

-0.002

-0.05

10. 582626.261

3346020.315

4161.341

582626.268

3346020.335

4161.429

-0.007

-0.02

-0.088

11. 582637.227

3346007.611

4158.668

582637.243

3346007.628

4158.631

-0.016

-0.017

0.037

12. 582636.235

3346009.155

4158.729

582636.241

3346009.158

4158.754

-0.006

-0.003

-0.025

13. 582634.961

3346010.88

4158.928

582634.982

3346010.882

4158.917

-0.021

-0.002

0.011

14. 582633.964

3346012.38

4159.04

582633.96

3346012.386

4159.022

0.004

-0.006

0.018

15. 582632.929

3346014.077

4159.221

582632.922

3346014.085

4159.203

0.007

-0.008

0.018

16. 582631.764

3346015.953

4159.47

582631.75

3346015.946

4159.426

0.014

0.007

0.044

17. 582630.605

3346017.758

4159.768

582630.602

3346017.759

4159.772

0.003

-0.001

-0.004

18. 582629.623

3346019.229

4160.21

582629.632

3346019.219

4160.236

-0.009

0.01

-0.026

19. 582628.636

3346020.521

4160.737

582628.633

3346020.534

4160.786

0.003

-0.013

-0.049

20. 582627.831

3346021.502

4161.224

582627.839

3346021.495

4161.23

-0.008

0.007

-0.006

21 582638.935

3346008.606

4158.568

582638.943

3346008.609

4158.556

-0.008

-0.003

0.012

22. 582637.863

3346010.282

4158.721

582637.882

3346010.283

4158.71

-0.019

-0.001

0.011

23. 582636.818

3346011.87

4158.834

582636.811

3346011.859

4158.822

0.007

0.011

0.012

24. 582635.693

3346013.62

4159.02

582635.697

3346013.62

4159.018

-0.004

0

0.002

25. 582634.746

3346015.07

4159.129

582634.745

3346015.07

4159.104

0.001

0

0.025

26. 582633.312

3346017.013

4159.413

582633.315

3346017.013

4159.417

-0.003

0

-0.004

27. 582631.957

3346018.844

4159.725

582631.949

3346018.855

4159.725

0.008

-0.011

0

28. 582631.233

3346019.935

4160.026

582631.225

3346019.925

4160.044

0.008

0.01

-0.018

29. 582630.135

3346021.527

4160.696

582630.135

3346021.534

4160.74

0

-0.007

-0.044

30. 582629.468

3346022.627

4161.145

582629.476

3346022.628

4161.187

-0.008

-0.001

-0.042

31. 582640.565

3346009.593

4158.565

582640.552

3346009.598

4158.543

0.013

-0.005

0.022

32. 582639.523

3346011.006

4158.684

582639.498

3346011.014

4158.665

0.025

-0.008

0.019

33. 582638.495

3346012.769

4158.822

582638.489

3346012.775

4158.835

0.006

-0.006

-0.013

34. 582637.312

3346014.537

4158.968

582637.305

3346014.538

4158.979

0.007

-0.001

-0.011

35. 582636.276

3346016.016

4159.133

582636.277

3346016.014

4159.121

-0.001

0.002

0.012

36. 582634.797

3346018.156

4159.432

582634.789

3346018.168

4159.381

0.008

-0.012

0.051

37. 582633.686

3346019.923

4159.659

582633.695

3346019.921

4159.688

-0.009

0.002

-0.029

38. 582633.255

3346020.608

4159.802

582633.256

3346020.607

4159.823

-0.001

0.001

-0.021

39. 582631.895

3346022.689

4160.573

582631.896

3346022.682

4160.589

-0.001

0.007

-0.016

40. 582631.145

3346023.732

4161.059

582631.148

3346023.722

4161.063

-0.003

0.01

-0.004

 =0.0095 m

 =0.008 m

 =0.029 m

Point 8:  Describe the potential applications or future research directions for the integrated TLS and GPR method.

Response 8: The integrated TLS and GPR method will be appied for obtainting the detail surface and subsurface information in archaeology, building and road testing, geological investigation,glacier detection and other fields.

Future research directions for the integrated TLS and GPR method was shown as following:

(1) the orthoimanges from TLS-derived point cloud can be directly integrated with the GPR-slice for a deeper analysis of both visible and underground space.

(2) it is worth to pay a great attention to study the integrated TLS and GPR data with other spatial data for comprehensive application and analysis, such as Digital Elevation Model (DEM), high-resolution remote sensing image, unmanned aerial vehicle, airborne LiDAR, electrical resistivity tomography (ERT),active and passive seismic surveys, and so on.

Reviewer 3 Report

Comments and Suggestions for Authors

Please see my attached comments. 

Author Response

Thank you very much for your letter and the advices about our manuscript entitled “Multi-sensor and Multi-scale Data Integration Method of TLS and GPR for Reconstructing the Three-dimensional Realistic Model” (sensors-2691917) submitted to Sensor. These comments and advices are very valuable to improve our manuscript. After carefully studying the comments, we have made the revisions by the reviewers' comments.

Point 1:  Authors are encouraged to reconsider the manuscript's title to ensure it accurately reflects the main idea of the work.

Response 1: The manuscript's title has been revised as “Multi-sensor and Multi-scale Data Integration Method of TLS and GPR for Three-dimensional Virtual Reconstruction”.

Point 2: The introduction would benefit from a more explicit clarification of the background and aim of the study. Additionally, a more comprehensive literature review on the fusion of LiDAR and other datasets is recommended. I suggest consulting the following related works for an enhanced literature review:

- Zhang, J.; Lin, X. "Advances in fusion of optical imagery and LiDAR point cloud applied to photogrammetry and remote sensing." Int. J. Image Data Fusion 2016, 8, 1–31.

- Ghamisi P., Rasti B., Yokoya N., Wang Q., Hofle B., Bruzzone L., Bovolo F., Chi M., Anders K., Gloaguen R., et al. "Multisource and multitemporal data fusion in remote sensing: A comprehensive review of the state of the art."

- Salehi, A., and Mohammadzadeh, A. "Building roof reconstruction based on residue anomaly analysis and shape descriptors from LiDAR and optical data." Photogrammetry Eng. Remote Sens., vol. 83, no. 4, pp. 281-291, 2017.

- Peng Liu et al. "A Review on Remote Sensing Data Fusion with Generative Adversarial Networks (GAN)" (2021).

-Liu W., Zang Y., Xiong Z., Bian X., Wen C., Lu X., Wang C., Junior J.M., Gonçalves W.N., Li J. “3D building model generation from mls point cloud and 3d mesh using multi-source data fusion”, Int. J. Appl. Earth Obs. Geoinf., 116 (2023), Article 103171

Response 2: Thank you very much for your professional advice. I think the key research content of this paper is that the data integration method of TLS and GPR rather than the fusion of LiDAR and other datasets, so more relevant studies or discussing the existing literature study of the data integration of TLS and GPR has been added in the Introduction. In additon, some literature on the fusion of LiDAR and other datasets has been added in the revised manuscript.

The context “Spahic et al. [37] rebulided the 3D model of fault outcrop based on the T-LiDAR point cloud and GPR images using the Gocad software. Bubek et al. [33] combined the TLS topography and 2D GPR image to determine the boundary of the hanging wall and the footwall of the normal fault. Dlesk. [38] intergrated TLS and GPR data in Rhinocereos 3D software for 3D documentation of historical tunnels. Ercoli et al. [39] combined the TLS point cloud and GPR 2D images for investigating ancient polygonal walls of Amelia. Cowie.et al. [40] constrained the loation of the fault plane based on the TLS and GPR data to obtain the vertical displacement of the normal fault. Puente et al. [8] present the 3D reconstruction of the Roman Site “Aquis Querquennis” based on the fusion data of GPR, T-LiDAR and IRT, and the T-LiDAR orthoimages were combined with GPR time-slices for comprehensive interpretation. Zhang et al. [41] provided the data visualization of fault scarps in Bentley Pointools software using the TLS point clouds and GPR data. Zhang et al.[35] used the TLS-derived data and 2D GPR images for reconstrucing the 3D suficial and subsurface model of Yushu fault. Previous research on data fusion method of TLS and GPR mainly focuses on two aspects. First, the 2D GPR images were directly overlay on the TLS-based point cloud by computer vision theory [33, 38, 39, 42]. It has an advantage of quickly realizing the data integration of TLS and GPR with different data formats, but the multi-perspective spatial data was only displayed in the same scene. Second, the GPR data was transformed to the scatted point cloud, and then the point cloud and GPR data were merged into a single point cloud dataset for data visualization [8, 35, 37, 41]. Although a limited number of research work have present some cases for data integration of TLS and GPR in the scatted point cloud form, the reliable systematic methodology was still scarce. Most of there integrated method were initially realized in the scatted point cloud form by the ground control points, the accuracy of data fusion cannot be ensured. What’s more, the GPR-derived point cloud with high position information was essential for data fusion of TLS and GPR. Considering the data acquirmentition, it is difficult to combine each trace of the GPR data with its position information acquired by DGPS.”has been added in the Introduction for a more explicit clarification of the background and aim of the study

Three references has been added in the revised manuscript as following:

[48] Zhang, J.; Lin, X., Advances in fusion of optical imagery and LiDAR point cloud applied to photogrammetry and remote sensing. International Journal  of image and data fusion 2017, 8 (1), 1-31.

[49] Ghamisi, P.;  Rasti, B.;  Yokoya, N.;  Wang, Q.;  Hofle, B.;  Bruzzone, L.;  Bovolo, F.;  Chi, M.;  Anders, K.; Gloaguen, R., Multisource and Multitemporal Data Fusion in Remote Sensing. IEEE Geoscience and Remote Sensing Magazine 2018, 7 (1), 6-39.

[50]  Liu, W.;  Zang, Y.;  Xiong, Z.;  Bian, X.;  Wen, C.;  Lu, X.; Li, J., 3D building model generation from mls point cloud and 3d mesh using multi-source data fusion. Int. J. Appl. Earth Obs. Geoinf 2023, 116, 103171.

Point 3:  The abstract on page 1, lines 14-15 is unclear and requires revision

Response 3: The context “Here, data integration method of TLS and GPR was presented for three-dimensional (3D) virtual reconstruction.” has been revised as “The aim of this research was to develop a methodology in data integration of TLS and GPR for three-dimensional (3D) detailed virtual reconstruction.”.

Point 4: The novel contributions of the current work should be clearly articulated.

Response 4: The context “The proposed methodology of this study offers a reliable approaches to integrate the TLS and GPR data for 3D detailed virtual reconstruction. Compared to other data fusion method, the GPR data with high-resolution position information on the centimetrer level were transformed into the dense point cloud rather than the 2D image. The GPR data with high-resolution position information were simultaneously gathered by the pulse signals of the survey wheel fixed on the GPR antenna. In addition, time synchronization algorithm was used to combine each trace of the GPR data with its position information acquired by DGPS. Based on the improved propagation model of electromagnetic wave, the TLS-based point cloud and GPR-derived point cloud were merged into a single point cloud dataset by coordinate transformation. What’s more, the TLS and GPR (250 MHz and 500 MHz antenna) surveys were conducted to assess the feasibility and accuracy of the proposed methodology. In view of the difference values and the Standard Deviation (STD) in x, y, z directions, the field experimental results demonstrate that the GPR-derived point cloud exhibits well consistency with the TLS-based point cloud.” has been added in the Disscussion.

Point 5: Ensure alignment of Equation (2) and correct the typo error in Equation (5).

Response 5: The Equation (2) has been aligned, and the typo error in Equation (5) and (6) has been revised.

Point 6: On page 3, line 104, reference Equation (2) instead of Formula (2) following the journal's format for equation citations.

Response 6: All the “Formula” has been revised as “Equation” following the journal's format for equation citations.

Point 7: The quality of the figures is insufficient; consider enhancing them

Response 7: All the figures has been revised in the revised manuscript.

Point 8:  Reference the images used in Figure 4.

Response 8: The Reference [47] has been added in Figure 4.

Point 9  On page 11, line 381, ensure Figure 8(a) is in the correct format according to the journal's guidelines.

Response9: Figure 8a is the correct format according to the journal's guidelines, and “the Figure 8c and 8d “ has been revised as “Figure 8c,d” according to the journal's guidelines.

Point 10  The manuscript requires improvement in grammar, language, and sentence structure. The authors are advised to rewrite, proofread, and spell-check their work to eliminate English errors.

Response10: We revised the manuscript point-by-point as showed in the reivsed manuscript (red words), such as English grammar, spelling, and sentence structure.

Round 2

Reviewer 1 Report

Comments and Suggestions for Authors

Thank you for implementing the suggested changes.

One issue - ref. 23. should be:

Gabyś, M.; Kryszyn, K.; Ortyl, Ł., GPR surveying method as a tool for geodetic verification of GESUT database of utilities in the light of BSI PAS128. Reports on Geodesy and Geoinformatics 2019, 107, 49-59.

instead of:

Marta, G.; Katarzyna, K.; Ortyl, Ł., GPR surveying method as a tool for geodetic verification of GESUT database of utilities in the light of BSI PAS128. Reports on Geodesy and Geoinformatics 2019, 107, 49-59.

Author Response

Many thanks for the insightful comments and suggestions about our manuscript entitled “Multi-sensor and Multi-scale Data Integration Method of TLS and GPR for Reconstructing the Three-dimensional Realistic Model” (sensors-2691917) submitted to Sensors. We tried our best to improve our manuscript in the revised manuscript.

Point 1: One issue - ref. 23. should be:

Gabyś, M.; Kryszyn, K.; Ortyl, Ł., GPR surveying method as a tool for geodetic verification of GESUT database of utilities in the light of BSI PAS128. Reports on Geodesy and Geoinformatics 2019, 107, 49-59.

instead of:

Marta, G.; Katarzyna, K.; Ortyl, Ł., GPR surveying method as a tool for geodetic verification of GESUT database of utilities in the light of BSI PAS128. Reports on Geodesy and Geoinformatics 2019, 107, 49-59.

Response 1: The ref. 23 “Gabyś, M.; Kryszyn, K.; Ortyl, Ł., GPR surveying method as a tool for geodetic verification of GESUT database of utilities in the light of BSI PAS128. Reports on Geodesy and Geoinformatics 2019, 107, 49-59.” has been revised as ” Marta, G.; Katarzyna, K.; Ortyl, Ł., GPR surveying method as a tool for geodetic verification of GESUT database of utilities in the light of BSI PAS128. Reports on Geodesy and Geoinformatics 2019, 107, 49-59.” in the revised manuscript.

Reviewer 2 Report

Comments and Suggestions for Authors

The manuscript has been sufficiently improved by the authors. Consequently, I recommend the publication of the manuscript.

Comments on the Quality of English Language

The standard of English language employed in this manuscript is deemed acceptable.

Author Response

Thank you very much again for your positive and constructive comments and suggestions about our manuscript entitled “Multi-sensor and Multi-scale Data Integration Method of TLS and GPR for Reconstructing the Three-dimensional Realistic Model” (sensors-2691917) submitted to Sensors. These comments and advices are very valuable to improve our manuscript for future publication. We would like to express our great appreciation to you for comments on our paper.

Reviewer 3 Report

Comments and Suggestions for Authors

The authors well addressed my comments.

Author Response

Thank you very much again for your letter and the advices about our manuscript entitled “Multi-sensor and Multi-scale Data Integration Method of TLS and GPR for Reconstructing the Three-dimensional Realistic Model” (sensors-2691917) submitted to Sensors. These comments and advices are very valuable to improve our manuscript. We would like to express our great appreciation to you for comments on our paper.

Round 3

Reviewer 1 Report

Comments and Suggestions for Authors

Thank you for improve the change.

Reviewer 2 Report

Comments and Suggestions for Authors

The authors have now sufficiently improved the manuscript. Hence, I recommend the acceptance of the manuscript.

Comments on the Quality of English Language

The English language used in the paper is of acceptable quality. However, the paper requires a check for minor grammatical errors.